# Semantic Communities from Graph-Inspired Visual Representations of Cityscapes

**Vasiliki Balaska** [1] **, Eudokimos Theodoridis** [1] **, Ioannis-Tsampikos Papapetros** [1] **, Christoforos Tsompanoglou** [1] **, Loukas Bampis** [2] **and Antonios Gasteratos** [1,*]

1   Department of Production and Management Engineering, Democritus University of Thrace, Vas. Sophias 12, 671 32 Xanthi, Greece
2   Department of Electrical and Computer Engineering, Democritus University of Thrace, University Campus Kimmeria, 671 00 Xanthi, Greece
*   Correspondence: agaster@pme.duth.gr; Tel.: +30-2541-079359

**Abstract:** The swift development of autonomous vehicles raises the necessity of semantically mapping the environment by producing distinguishable representations to recognise similar areas. To this end, in this article, we present an efficient technique to cut up a robot's trajectory into semantically consistent communities based on graph-inspired descriptors. This allows an agent to localise itself in future tasks under different environmental circumstances in an urban area. The proposed semantic grouping technique utilizes the Leiden Community Detection Algorithm (LeCDA), which is a novel and efficient method of low computational complexity and exploits semantic and topometric information from the observed scenes. The presented experimentation was carried out on a novel dataset from the city of Xanthi, Greece (dubbed as $Gryphon_{urban}$ urban dataset), which was recorded by RGB-D, IMU and GNSS sensors mounted on a moving vehicle. Our results exhibit the formulation of a semantic map with visually coherent communities and the realisation of an effective localisation mechanism for autonomous vehicles in urban environments.

**Keywords:** community detection; semantic segmentation; semantic mapping; graph-inspired descriptors; $Gryphon_{urban}$ dataset; autonomous navigation





## 1. Introduction

Contemporary research provides modern autonomous systems with the ability to semantically recognise and categorise regions or entities, as presented in [1,2]. An autonomous system is one that can achieve a given set of goals in a changing environment—gathering information about its surroundings and working for an extended period of time without human control or intervention. Driverless cars and autonomous mobile robots (AMRs) used in warehouses are two common examples [3]. In this regard, the interpretation of complex environments, such as urban scenes, constitutes a challenging task, with disturbances from dynamic or obscured entities, leading to a growing interest in the semantic segmentation and mapping of urban environments [2,4].

In addition, during the navigation of a vehicle within a residential area, entities that contain important clues for humans should be detected and recognised. Furthermore, Artificial Intelligence (AI) enhances the autonomous agents' capabilities to semantically perceive their environment and accurately recall spatial memories, contributing to the fundamentals of cooperation between humans and robots or among multiple agents. Therefore, autonomous platforms must maintain the cognitive ability to interpret locations and extract semantic information to represent their environment correctly. The most appropriate way to organize all this information relies on a semantic map, which corresponds to an enhanced representation of the environment with high-level geometric information and quality features.

However, due to the dynamics and plethora of different entities in an urban scene, the formulation of semantic communities requires thorough research, as it is currently lacking in the field. In this paper, our methodology is visualised in Figure 1, establishing the creation of undirected graphs with semantic and metric information to describe an urban scene, contributing to the above segmentation. Specifically, our main contributions that are important to the research community are presented below:

- A robust model to generate semantic communities in an urban challenging environment based on a community detection algorithm of graph-inspired topometric descriptors of observed entities;
- The creation of graph-based description vectors, for which we semantically segment every input image and produce the corresponding descriptor in the form of an undirected graph;
- A novel dataset recorded in the city of Xanthi, Greece, with a moving car that contains distinct semantic regions with consistent visual information, in order to validate our system.

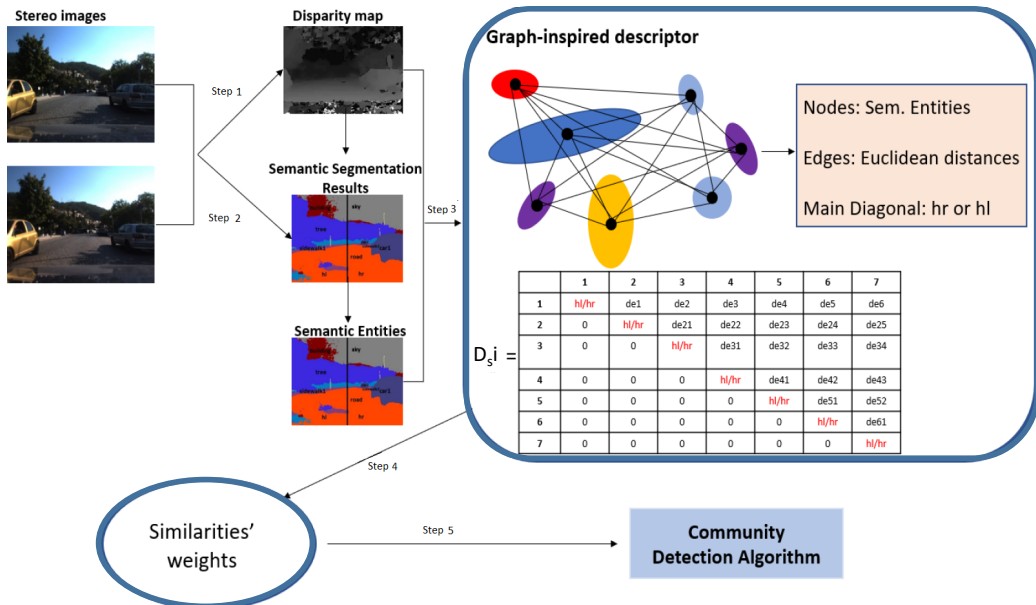

**Figure 1.** Schematic representation of the procedural steps for the proposed system. Firstly, each input image is semantically segmented, and a disparity map for each stereo image pair is computed. Then, communities are detected by means of graph-based descriptors that include nodes representing the designated entities and edges weighted by the *L*2-score between their centroids.

The rest of the paper is structured as follows. In Section 2, we discuss representative works from the related literature. Section 3 contains a detailed explanation of our approach. In Section 4, we present the procedure followed in order to form our novel $Gryphon_{urban}$ dataset from the city of Xanthi, Greece, and we present the results of our experiments. In Section 5, a useful discussion of our findings is provided, while in Section 6, we draw conclusions and present our plans for future work.

## 2. Related Literature

In this section, two main different mapping approaches are utilised. The first one is metric maps, retaining geometric information about the environment, while the second one, namely the topological maps, represents the surroundings in the form of a graph containing a set of regions interconnected with links. The combination of the above models

produces the so-called "topometric map" [5,6]. The first step in creating a topometric map of the environment requires the calculation of similarity measurements between successive camera measurements. The mechanisms for quantifying these similarities, in terms of the degree to which information is extracted, vary between different approaches; however, three categories can be identified, namely image-based, histogram-based and Bag-of-Words (BoW). Image-based techniques use pixel differences between consecutive images to detect any change in a scene. In histogram-based methods, the images are compared using feature statistics [7]. Finally, the Bag-of-Words model is the standard approach for simplifying visual representations and similarity statistics [8]. Topometric maps, by means of clustering algorithms, are utilised to categorize the different indoor regions in the corresponding clusters [9]. Besides, multiple sources of information can be exploited towards a robust topometric semantic map suitable for street-moving vehicles' trajectories [4]. The final map combines low-cost yet less accurate semantic information from satellite images with more detailed data from street-view camera measurements. The traditional definition of occupancy grids has been extended to that of a semantic occupancy grid [10], which encodes the presence or absence of an object category at each grid location. The main objective of such a representation is to predict the probability that each semantic class presents at each location in a topometic birds-eye-view map [11]. In addition, other works propose and evaluate a learning-based architecture to perform semantic segmentation, place categorisation and topometric mapping on occupancy grid maps [12].

A semantic map provides an enriched description of the environment intended for advanced navigation, fine planning and robust localisation [1,13]. Besides, semantic maps have been proposed as an information tool for land planning and pasture management [14]. Existing works [15,16] have revealed that high-level semantic features provide a more robust representation of the scene as they incorporate the information of object properties and their mutual relationships. Hence, they are able to successfully deal with global localisation under extreme appearance changes [15]. Lightweight mapping methods have been proposed to organize environmental objects' semantic and geometric properties through a topological graph. On that basis, a robust localisation based on the graph description of the semantically segmented local scene is built, called an object-level Topological Semantic Map (OLTSM) [17]. Some additional works, such as [18,19], utilise sequential data throughout a trajectory to identify different regions within it, while semantic place partitioning is achieved through a learned vocabulary combined with both spatial and temporal information [20]. Moreover, a quad-tree method to decompose the environment into cells and a spectral clustering technique to group them into semantic regions is presented in [21]. The identification of intersections and terminal points within an occupancy grid provides the means for segmenting metric maps into semantic regions [22]. The method proposed in [23] was based on the Single-Linkage (SLINK) agglomerative algorithm [24]. This unsupervised and incremental approach allows an autonomous system to learn about the organisation of the observed environment and localisation within it. Lastly, Census Transform Histogram (CENTRIST) and GIST descriptors were used to apply an unsupervised approach with a Self-Organising Map for classification [25]. Essentially, these approaches are based on detecting distinct segments in the robot's trajectory by exporting local features from each input image in order to achieve localisation with respect to the recognised regions. Nevertheless, even though their effectiveness was proven for indoor environments, they have not been tested in more dynamic urban scenes (containing many dynamic entities such as pedestrians, moving vehicles, traffic lights, changing terrain, etc.). Therefore, the successful formulation of semantic communities in urban areas is still an open challenge. Creating graph-based descriptors is an innovative solution to achieve accurate outdoor localisation, and [26,27] were based on the created graphs, achieving high performance.

## 3. Approach

In this section, our approach to creating semantic communities in an outdoor environment is detailed by describing the selected technique for representing the input images,

as well as the algorithm for clustering the feature vectors according to their similarity. Unlike other information mechanisms, such as visual word histograms, the proposed graph-inspired descriptor leverages semantic and topometric information from the environment. Hence, the first step of the proposed method is the semantic recognition of entities and the computation of their full pose relative to the world coordinate frame. Then, the above information is used to generate representative descriptors from each input image, fed into a community detection approach to generate a semantic map of the robot's route. To evaluate our method, we created an image dataset from a vehicle's trajectory in the streets of the town of Xanthi.

*3.1. Generation of Graph-Based Descriptors*

In order to achieve the semantic segmentation of the obtained images while emphasising high applicability and low computational complexity, we adopt the open-source SegNet [28] architecture, which is widely acknowledged in such applications [29]. This network is pre-trained on the CamVid dataset [30] on the following class labels: *sky, building, road, tree, side-walk, pole, road marking, bicyclist, sign symbol, car, person* and *unlabelled*. The trained SegNet model is utilised to infer our $Gryphon_{urban}$ dataset, achieving a mean Intersection of Union (mIoU) accuracy (inference accuracy) of 62%. Throughout, the input of this network is a $360 \times 480$ frame; thus, each recorded image is converted to the respective size. Furthermore, discrete entities from each semantic class are detected by applying the Moore–Neighbour detection algorithm [31], modified by Jacob discontinuity criteria, over the segmentation outcome, resulting in a set $E$ of $e$ different entities. The semantic results of the $Gryphon_{urban}$ imagery indicate that a maximum of two entities per semantic class is located in every scene. Additionally, in order to incorporate topological information into our proposed descriptors, each processed image is divided into two parts, namely the right ($r$) and the left ($l$), as shown in Figure 1. Thus, each of the detected entities is assigned to one of the above sections, based on the majority of its pixels. The road class is constantly assigned to $r$ since no more than one instance per frame is detected for the whole dataset.

The process of localising semantic entities from a running vehicle requires the extraction of depth information and the 6DoF position of the respective camera sensor. Depth data can be obtained via various approaches, such as the ones in [32,33]. Our visual imagery dataset was created by utilising a basic set-up comprising a stereo camera rig and a GNSS sensor, associated with the recorded image stream. Therefore, due to its low computational complexity, we compute the disparity measurements based on the Semi-Global Matching (SGM) method due to its low computational complexity [34]. With the above information, the position of each semantic entity is calculated according to the procedure described in [34].

As the robot explores a new environment, each scene is described through metric and semantic information. Specifically, each image captured from the robot's path is interpreted in the form of an undirected graph. Therefore, for each semantic entity in every scene, we compute its centroid in the 3D world ($c_{me}$). Graph nodes represent semantic entities ($c_{me}$), while edges are weighted by the Euclidean distance between them. Thus, the proposed descriptor includes $k = 2 \cdot e$ nodes, while the multitude of edges is equal to $\frac{k \cdot (k-1)}{2}$. In addition, each node receives a binary value (0 or 1), depending on its topological position in relation to the vehicle (left or right, respectively). Figure 1 exhibits the resulting graph for each street-view frame, which can be interpreted as a square matrix $D_s$. The values of the main diagonal of $D_s$ denote the aforementioned topological position. Each descriptor is retained as a sparse matrix to account for the fact that the existence of each class is not always guaranteed, ensuring memory efficiency. Finally, the description vectors are created by a vectorisation process. Algorithm 1 describes the above procedure for generating graph-inspired description vectors ($D_v$).

The next step of our approach refers to the calculation of similarity values among the recorded images through the computed $D_v$. Specifically, we base this procedure on the similarity $L2$-score [35]:

$$W = 1 - 0.5 \cdot ||\bar{D}_{v1} - \bar{D}_{v2}||_2, \tag{1}$$

where $\bar{D}_{v1}$ and $\bar{D}_{v2}$ denote $L2$ normalised unit vectors. This normalisation is applied to constrain the effect of certain regions containing significantly more semantic entities than others. Score $W \in [0, 1]$ quantifies the similarity among graph-based description vectors $\bar{D}_v$, with higher values indicating increased semantic and topometric correlation. Subsequently, we shape a similarity matrix $M$ quantifying the degree of resemblance among all recorded frames, which is further utilised to deduce whether the related images belong to the same community.

---

**Algorithm 1:** Pseudocode algorithm for the creation of the proposed graph-based description vector.

---

1   **Input**: $A$: $l$ or $r$ labels and $e$: number of entities
2   **Output**: $D_v$ description vector
3   **for** $t_1 = 1, 2, \ldots e$ **do**
4      **if** $A(t_1) == 'l'$ **then**
5         $D_s[t_1, t_1] \leftarrow 0$
6      **else**
7         $D_s[t_1, t_1] \leftarrow 1$
8      **end if**
9      **for** $t_2 = t_1, 2, \ldots e$ **do**
10         $d_e \leftarrow ||cm_{e(t_1)} - cm_{e(t_2)}||_2$
11         $D_s[t_1, t_2] \leftarrow d_e$
12      **end for**
13   **end for**
14   $D_v \leftarrow vec(D_s)$ // vectorisation of $D_s$

---

### 3.2. Generation of Semantic Communities

Matrix $M$ contains the information regarding the complete similarity structure of the executed route. At this point, we need to emphasise that our goal is to group coherent trajectory regions. Thus, the whole matrix is converted into a new graph, the nodes of which correspond to images and the edges to their similarity scores. Then, clustering is achieved through the LeCDA [36] due to its thoroughly explored communities, which guarantees strong connections among the contained nodes. In our previous works [9,37], a predecessor of LeCDA was utilised for a similar task, namely the Louvain Community Detection Algorithm (LaCDA) [38]. However, as described in [36], the LaCDA may lead to disconnected communities, where two or more clusters co-exist in a community without sharing any edges. To this end, instead of directly aggregating nodes to a new level, Leiden integrates a refinement phase to evaluate the consistency of the formed communities. In addition, adopting a fast local move algorithm for node movement among the communities contributes to speeding up the whole process. The crucial objective of both algorithms is to achieve maximum modularity, which is used to measure the strength of dividing a network into communities. The modularity is calculated as follows:

$$H = \frac{1}{2m} \sum_c {}^1 (e_c - \gamma \frac{K_c^2}{2m}), \tag{2}$$

where $e_c$ is the number of edges in the community $c$, $m$ is the total number of edges in the network, $\gamma > 0$ is a resolution parameter that expresses the number of communities that will be formed and $K_c$ is the sum of the edges in the community $c$. According to [36], $\gamma > 0$ is a resolution parameter, where higher resolutions lead to more communities, while lower

resolutions lead to fewer communities. In our work, we based the selection of $\gamma$ on the work of [39], who analysed the Hamiltonian function and its different parameters. Each node belongs to the community that yields the maximum modularity value. This phase is repeated until local maximum modularity is achieved, and then, all the communities are reduced to vertices, creating a new graph. By means of LeCDA, the images are clustered into multiple community configurations, each of which is associated with a particular modularity value. We choose maximum modularity to produce the best possible segmentation.

## 4. Results

In this section, our approach is extensively evaluated for the accuracy of the experimental setup that makes up the proposed technique through the $Gryphon_{urban}$ dataset. Secondly, comparative results are presented among the proposed graph-based descriptors, with different similarity metrics and SURF, SIFT and ORB features for the two community detection algorithms.

### 4.1. $Gryphon_{urban}$ Dataset Formulation

In order to effectively evaluate our system, samples from an outdoor environment are required with distinctive semantic regions. To achieve that, we recorded a novel dataset from an urban area located in Xanthi. The $Gryphon_{urban}$ dataset (https://robotics.pme.duth.gr/research/gryphon_urban/ (accessed on 9 September 2022)) was recorded from a moving vehicle while driving around the city, obtaining a few thousand frames. We implemented one route within the city during the morning of a sunny day, resulting in about 1500 street-view images. Our camera system and post-processing reflect the current state-of-the-art in the automotive domain. RGB images were recorded with an automotive-grade 12 cm baseline stereo camera and a focal length of 2.5 mm with 97° HFOV. Each stereo image pair was subsequently debayered and rectified. We relied on [40] for extrinsic and intrinsic calibration. The resulting images were visually more pleasing and proved easier to annotate. The frame rate of the camera is 48 fps. This dataset contains 8868 unlabeled images captured at three different times in a day. Moreover, 291 semantic labeled images were randomly selected from the dataset. The size of each image file is $640 \times 480$ pixels, while the file is associated with geolocation details. The Robotics Operation System (ROS) Melodic version implemented the image-capturing procedure. The trunk of our vehicle housed a Single Board Computer (SBC) with an Intel Core 2 Duo processor, a shockproof hard disk with a storage capacity of 250 GB and a smartphone with the Share GPS application installed. Every image in our dataset is assigned its respective vehicle velocity and GNSS data, recorded by the Share GPS application and synchronised through ROS. The traversed route was 10 km, and the execution time was calculated at 45 min with an average vehicle velocity of 40 km/h. Figure 2 visualises the route's overview, highlighted with a blue line.

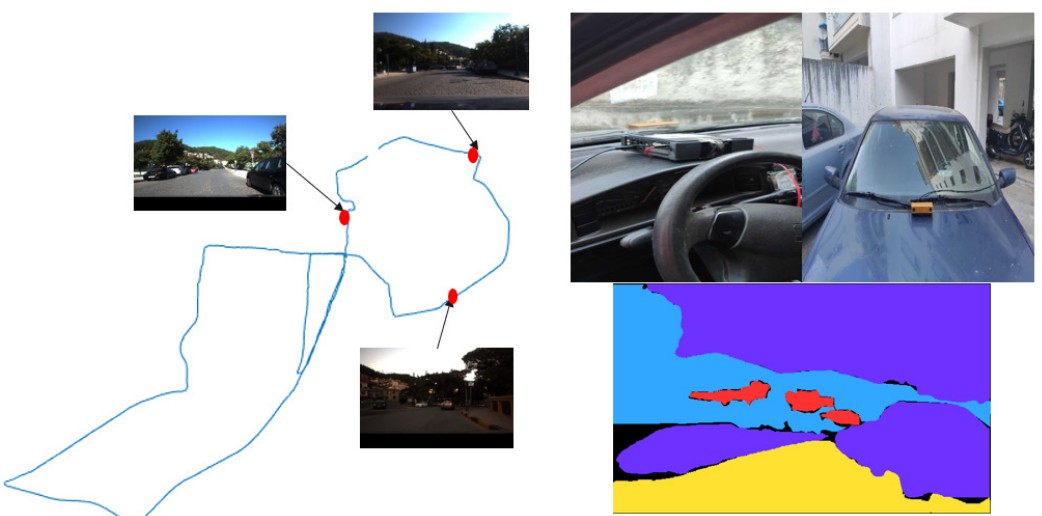

**Figure 2.** Visualisation of the proposed *Gryphon<sub>urban</sub>* dataset, together with the hardware set-up and vehicle used for its creation. Additionally, some images from the route with the corresponding semantic information are presented.

*4.2. Performance Evaluation*

First, our final system was tested in relation to the achieved performance of community detection. The selected measurement refers to the percentage of robot observations assigned to the proper community according to a manually created ground-truth map (see Figure 4, semantic areas 1, 2, 3 and 4). In order to conclude the proposed clustering process, several testing scenarios were performed. Specifically, the choice of the LeCDA was compared against the LaCDA one, evaluating the robustness of the exported semantic communities. The modularity value of LaCDA $\Delta Q$ is computed for all neighbouring communities:

$$\Delta Q = \left( \frac{\sum_{in} + k_{i,in}}{2m} - \left( \frac{\sum_{tot} + k_i}{2m} \right)^2 \right) - \left( \frac{\sum_{in}}{2m} - \left( \frac{\sum_{tot}}{2m} \right)^2 - \left( \frac{k_i}{2m} \right)^2 \right), \tag{3}$$

where $\sum_{in}$ is the sum of the links' weights within the community to which the node *i* is assigned, $\sum_{tot}$ is the sum of the links' weights associated with the nodes in the community, $k_{i,in}$ is the sum of the weights of the links from node *i* to the rest of the community, $k_i$ is the sum of the link's weights incident to node *i*, and *m* is the sum of the link's weights in the network.

In order to emphasise the robustness of the graph-based descriptors for creating semantic communities, further experiments were conducted related to the performance of the proposed description vectors' structure. Specifically, to finalise the form of graph-inspired descriptors, as analysed in Section 2, four different cases were developed:

*Case 1*: In this format, the proposed descriptor includes $k = 2 \cdot e$ nodes, while the multitude of edges is equal to $\left[ \frac{(k \cdot (k-1))}{2} \right]$ as discussed in Section 2. The value of each vector is assigned with 0 or 1, depending on the existence or absence of an entity. Therefore, no metric information is included among the semantic entities.

*Case 2*: This case is analysed in Section 3.1. The resulting size of the generated description vectors is similar to *Case 1*; however, the edges of the graph are weighted by the Euclidean distance among the entities' centroids.

*Case 3*: This case is similar to the above Case 2, except that we set each component of the main diagonal of the matrix to be equal to 0 in order to assess the effect of topological information in the description.

*Case 4*: In this case, the camera position is taken as the origin, and each centroid is assigned to a 3D vector from the origin. Similarly, the edges of the graph are weighted by the Euclidean distance among the entities and camera centroids.

Moreover, several types of similarity measurements for the generation of the corresponding weights, which are used in the respective community detection algorithm, were also evaluated: (i) *L*2-score, (ii) Sum of Absolute Differences (SAD) similarity and (iii) Jaccard index. Below, the SAD and Jaccard similarity measurements' formulas are presented, whilst for the *L*2-score, Equation (1) is applied in the converted unit vectors [41]:

$$S = 1 - \sum_{(i) \in D_v} |D_{v1}[i] - D_{v2}[i]|, \tag{4}$$

$$J = \frac{|\mathcal{D}_{v1} \bigcap \mathcal{D}_{v2}|}{|\mathcal{D}_{v1} \bigcup \mathcal{D}_{v2}|}, \tag{5}$$

where $D_{v1}$ and $D_{v2}$ represent the graph-inspired descriptors. For the computation of $J$, the $\mathcal{D}_v$ is the set of $D_v$ descriptor's values, which are converted to binary form, in which entities' existence is represented by 1, while no knowledge about their topology is included.

Table 1 shows the percentage of accurately clustered database instances to the proper community, as indicated by the ground-truth. More specifically, comparative results are presented among the *Case 1, Case 2, Case 3* and *Case 4* proposed graph-based descriptors, with different similarity metrics for the two community detection algorithms. As can be seen, the LeCDA combined with the *L*2-score in Case 2 achieves the highest accuracy, justifying our proposal for incorporating semantic, metric and topological information into a single descriptor. Figure 3 depicts the corresponding similarity matrices produced via score measurements from the most prominent graph-based descriptors' *Case 1 and Case 2* in comparison with *Case 3 and Case 4*. Finally, the performance of our proposal is compared against the corresponding accuracy of the SURF [42], SIFT [43] and ORB [44] descriptors. Note that we chose to compare with the ORB, SIFT, SURF algorithms, which, similar to our descriptor, do not require any training for their creation. Hence, for each case of the above-mentioned descriptors, we generate the corresponding BoW and a description histogram for every obtained camera frame. Then, we calculate the *W* scores among these vectors by applying Equation (1), and we create a similarity matrix to use within the LeCDA and LaCDA.

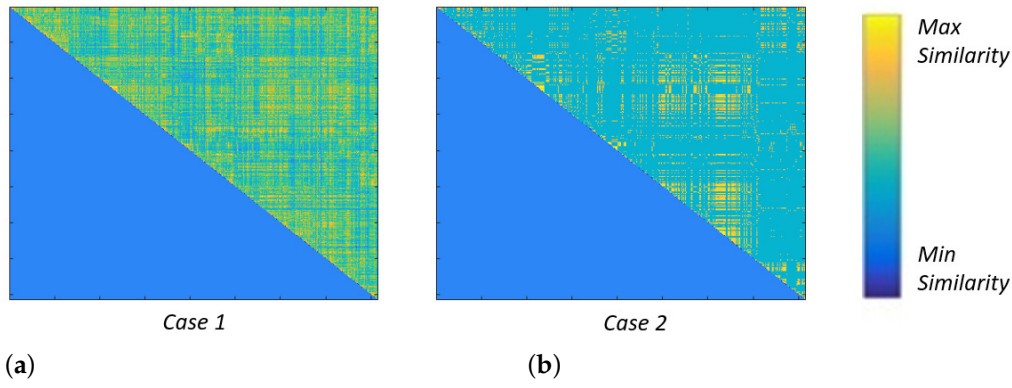

**Figure 3.** Similarity matrices produced through (**a**) Case 1 and (**b**) Case 2.

**Table 1.** Comparative accuracy (%) results of clustered ground-truth images in each semantic area for different experimental set-ups in *Case 1, Case 2, Case 3* and *Case 4*. Table entries marked with "-" correspond to communities that are not detected. The higher performance rates are represented in bold.

| Semantic Areas | | LeCDA | | | | LaCDA | | | |
|---|---|---|---|---|---|---|---|---|---|
| | | 1 | 2 | 3 | 4 | 1 | 2 | 3 | 4 |
| *Case 1* | *L*2-score | 25% | 67% | 60% | 51% | 34% | 56% | 62% | 49% |
| | Jaccard | 20% | 50% | 30% | 51% | 23% | 15% | 40% | - |
| | SAD | 31% | 75% | 62% | 32% | 22% | 79% | 58% | 45% |
| *Case 2* | *L*2-score | 55% | 100% | 92% | 71% | 42% | 60% | 74% | 56% |
| | Jaccard | 20% | 50% | 30% | 51% | 23% | 15% | 40% | - |
| | SAD | 49% | 88% | 76% | 32% | 31% | 85% | 67% | 48% |
| *Case 3* | *L*2-score | 20% | 59% | 60% | 50% | 34% | 56% | 62% | 45% |
| | Jaccard | 20% | 50% | 26% | 51% | 23% | 15% | 37% | - |
| | SAD | 27% | 73% | 60% | 30% | 18% | 75% | 55% | 39% |
| *Case 4* | *L*2-score | 22% | 65% | 60% | 51% | 30% | 50% | 60% | 49% |
| | Jaccard | 18% | 47% | 28% | 51% | 23% | 15% | 40% | - |
| | SAD | 30% | 75% | 60% | 31% | 20% | 77% | 58% | 45% |

Subsequently, Table 2 similarly shows the percentage of accurately clustered database instances to the proper community, as indicated by the ground-truth. More specifically, comparative results are presented among the proposed graph-based descriptors (*Case2*), with different similarity metrics and SURF, SIFT and ORB features for the two community detection algorithms. As can be seen, the LeCDA combined with the *L*2-score achieves high accuracy, justifying our proposal for incorporating semantic, metric and topological information into a single descriptor. On the contrary, hand-crafted features present low performance, as their discriminative capabilities are not sufficient to capture the complexity of an outdoor environment. Figure 4 visualises the semantic resulting maps for each experiment.

**Table 2.** Comparative accuracy (%) results of clustered ground-truth images in each semantic area for different experimental set-ups. Table entries marked with "-" correspond to communities that are not detected. Throughout, 0% performance indicates communities that are included but not recognised according to the ground-truth. The higher performance rates are represented in bold.

| Semantic Areas | | LeCDA | | | | LaCDA | | | |
|---|---|---|---|---|---|---|---|---|---|
| | | 1 | 2 | 3 | 4 | 1 | 2 | 3 | 4 |
| Ours (Case 2) | *L*2-score | 55% | 100% | 92% | 71% | 42% | 60% | 74% | 56% |
| | Jaccard | 20% | 50% | 30% | 51% | 23% | 15% | 40% | - |
| | SAD | 49% | 88% | 76% | 32% | 31% | 85% | 67% | 48% |
| SURF | | 20% | - | 20% | 0% | 22% | 0% | 15% | 0% |
| SIFT | | 49% | 20% | 32% | 20% | 41% | 10% | 36% | 18% |
| ORB | | 16% | 0% | 22% | 0% | 19% | 0% | 15% | 0% |

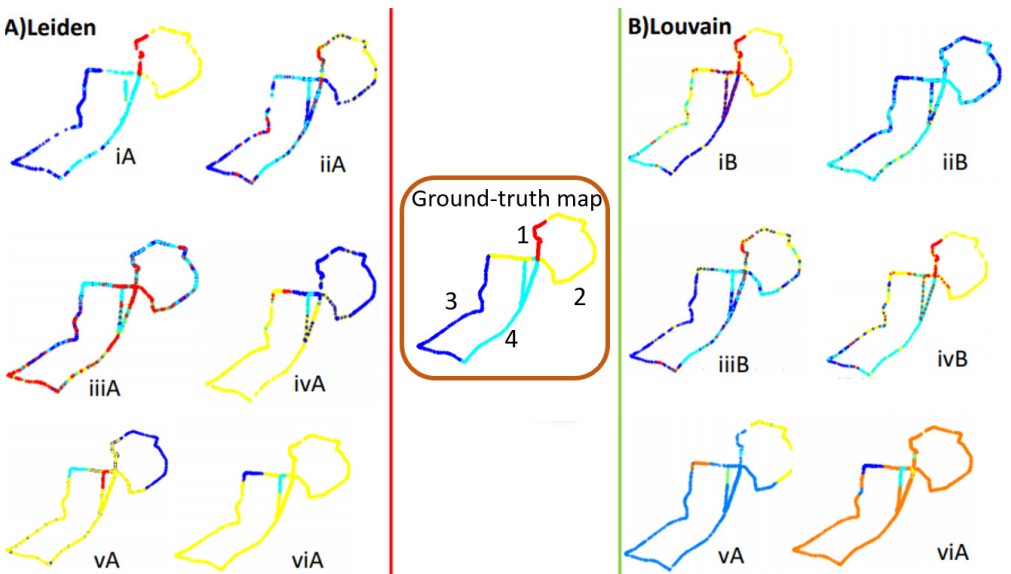

**Figure 4.** Mapping results produced for different community detection setups, as compared to the ground-truth. Cases (i), (ii) and (iii) arise from the implementation of graph-inspired descriptors, with (i) being produced through the $L2$-score, (ii) through the SAD-based similarity and (iii) through the Jaccard index. The semantic maps visualised in (ivA, ivB), (vA, vB) and (viA, viB) are extracted via SURF, SIFT and ORB descriptors, respectively.

## 5. Discussion

The ability to construct a map from a moving robot is essential for performing autonomous tasks and has been extensively studied according to the literature. Creating maps allows the robot to develop autonomous skills such as navigation, interaction with the environment and self-localisation, among others. The scientific community has focused on modern approaches to representing the environmental map in recent decades. Currently, the scientific community is becoming increasingly interested in so-called semantic solutions, which incorporate geometric information and semantic knowledge. Understanding the semantics of the human environment is the key to robots being truly integrated into everyday life. In general, ground-based robots that are capable of scheduling tasks usually combine semantic knowledge in their maps (e.g., categorising places, such as rooms, corridors, or gardens, and object labels). It is known that urban environments are especially challenging and require efficiency and robustness. The generation of semantic maps contributes to the solution of the above-mentioned problem, as their creation is based on the understanding of the environment by the robot, which resembles human comprehension. Therefore, with this work, we aim for every autonomous moving robot to recognize each area it encounters and to know the entities that exist in them so that it is able to move and localize correctly in a dynamic environment. To this end, it is necessary to properly organise knowledge in a methodological way so that it can be retrieved in a natural associative way by the robot, facilitating communication with humans or other agents. We hope that the creation of the proposed system will be one of the main applications of autonomous vehicles in the future. In general, in any case where GNSS is not available, the current system can be used to provide full localisation. For the creation of our proposed integrated system, initially, an unsupervised technique for semantic clustering and localisation was described. The available visual and odometry data were combined in Louvain or Leiden community detection algorithms to produce the topological map of a previously unexplored environment. However, environmental and road conditions can affect the created city-spaces. In this work, we made use of the part of the dataset that had been taken during the morning hours, considering that the lighting was sufficient for the entities' recognition. In order to reduce the effect of such conditions, the image segmentation mechanism needs to be trained over multiple environmental conditions in order to provide the necessary generalisation

capabilities. Another challenge remains with objects of the same category placed too close to each other. A possible solution to address those cases would be to adopt probabilistic methods that only accept correlations with high confidence and suspend these candidates until subsequent detections confirm their assignment.

## 6. Conclusions

In this article, a novel technique for generating semantic communities in outdoor challenging scenery is presented, based on the description properties we exploited through semantic and topometric information from a recorded scene. Our proposed descriptors achieve highly accurate and competitive results due to the realisation of the urban scenes into a graph-based representation of observed entities. Owing to their high fidelity and the inherited cityscape potentiality, the proposed descriptors are suitable for use in urban-related autonomous driving pipelines in order to confront complex applications, such as last-mile delivery. In our future work, we plan to expand our method into a condition-invariant representation by adopting image segmentation mechanisms capable of accommodating different lighting and environmental conditions to endow the method with the robustness required for the target applications.

**Author Contributions:** Conceptualization, V.B., L.B. and A.G.; methodology, V.B. and L.B.; software, E.T., I.-T.P. and C.T.; validation, V.B., E.T., L.B. and A.G.; formal analysis, V.B., E.T. and L.B.; investigation, E.T.; resources, A.G.; data curation, E.T.; writing—original draft preparation, V.B. and E.T.; writing—review and editing, L.B. and A.G; visualization, V.B. and E.T.; supervision, A.G.; All authors have read and agreed to the published version of the manuscript.

**Funding:** This research has been co-financed by the European Regional Development Fund of the European Union and Greek national funds through the Operational Program Competitiveness, Entrepreneurship and Innovation, under the call RESEARCH, CREATE, INNOVATE (project code: T2EDK-00592).

**Data Availability Statement:** Not applicable.

**Conflicts of Interest:** The authors declare no conflict of interest.

## Abbreviations

The following abbreviations are used in this manuscript:

| | |
|---|---|
| mIoU | Mean Intersection of Union |
| SLINK | Single Linkage |
| SAD | Sum of Absolute Differences |
| ROS | Robotic Operation System |
| SBC | Single Board Computer |
| SGM | Semi-Global Matching |
| LD | Census Transform Histogram |
| OLTSM | Topological Semantic Map |
| BoW | Bag-of-Words |
| LaCDA | Louvain Community Detection Algorithm |
| LeCDA | Leiden Community Detection Algorithm |

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
