# Peer review of "Semantic Communities from Graph-Inspired Visual Representations of Cityscapes"

_2673-4052, doi:10.3390/automation4010008_

Round 1

Reviewer 1 Report

The work presents a novel graph-based descriptor for semantic and topometric information in scenes. The authors build a semantic map using a community detection algorithm. The work evaluates the developed semantic mapping algorithm on a self-collected datasets with a vehicle driving through streets of a city. 

Grammar usage could use a bit of work. This greatly reduces readability and ease of understanding. The paper should be proofread very carefully.

The proposed approach of combining semantic, metric and topological information into a single descriptor is show to generate superior results for feature that are more distinctive for mapping. The approach is compared against feature descriptors such as ORB, SIFT, SURF. 

A comparison against deep learning based features is missing however. There has been significant research in learned feature descriptors both dense and sparse in the recent years. Some comparison against these learned descriptors would be quite helpful. 

The work does not evaluate the algorithm on benchmark datasets on localization and mapping such as KITTI, TUM-RGBD. 

It would be good to add some comparisons against existing semantic localization and mapping algorithms such as CubeSLAM, QuadricSLAM, by listing results on standard benchmark datasets used by the SLAM community.

Author Response

Dear Associate Editor and Reviewers,

With this cover letter, we are submitting the revised version of our manuscript entitled \textit{\textbf{"Semantic communities from graph-inspired visual representations of cityscapes"}}.

Thank you for the thorough review you have ensured for our manuscript and the valuable feedback provided. Please know that we have addressed all the raised comments and revised the manuscript accordingly. At this point, we wish to thank the anonymous referees for their constructive criticism, which allowed us to improve the quality of our paper. For this letter to be self-contained, we list each comment made by the reviewers and the associate editor, along with our specific responses. In addition, for your convenience, all significant changes have been highlighted in the revised manuscript with red color.

Yours sincerely,

Dr. Vasiliki Balaska

Reviewer \#1

\RC The work presents a novel graph-based descriptor for semantic and topometric information in scenes. The authors build a semantic map using a community detection algorithm. The work evaluates the developed semantic mapping algorithm on a self-collected datasets with a vehicle driving through streets of a city.

\RC Grammar usage could use a bit of work. This greatly reduces readability and ease of understanding. The paper should be proofread very carefully.

\AR  We acknowledge that our manuscript required thorough proofreading. We have corrected many grammatical and syntactical errors in our revised version.

\RC The proposed approach of combining semantic, metric and topological information into a single descriptor is shown to generate superior results for feature that are more distinctive for mapping. The approach is compared against feature descriptors such as ORB, SIFT, SURF. A comparison against deep learning based features is missing however. There has been significant research in learned feature descriptors both dense and sparse in the recent years. Some comparison against these learned descriptors would be quite helpful.

\AR We acknowledge the reviewer's suggestion for including additional experiments with features based on deep learning architectures. However, our intent is to provide some baseline comparative results with more traditional hand-crafted local features, such as, ORB, SIFT, SURF. In our future work, we plan to expand our proposal into a condition invariant representation by adopting image segmentation mechanisms capable of accommodating different lighting and environmental conditions. Therefore, we believe that the aforementioned deep learning approaches would be more suitable as comparative cases to highlight the feature extraction mechanism and not the total scene description, which is the main goal of the presented work.

Manuscript reference: In Section 4.2, second to last paragraph of revised paper, we state the reason for comparison with specific algorithms.

\RC The work does not evaluate the algorithm on benchmark datasets on localization and mapping such as KITTI, TUM-RGBD. It would be good to add some comparisons against existing semantic localization and mapping algorithms such as CubeSLAM, QuadricSLAM, by listing results on standard benchmark datasets used by the SLAM community.

\AR With the above comment of the reviewer, we seize the opportunity to clarify that two of our main contributions are the creation of the Dataset in which we applied our methodology and the extraction of graph-based descriptors. It should also be noted that TUM-RGBD is an indoor dataset, which could not be applied in this work. Furthermore, although KITTI corresponds to a widely used dataset, it does not include multiple traversals of the same areas. This essentially means that we cannot form the proposed semantic communities for a specific region and then perform localization using different memory and query sequences.

Most importantly, we wish to clarify that our work does not pose as a SLAM approach. On the contrary, it targets the general localization task where the robot's pose needs to be retrieved within a known environment using visual data. For this reason, we cannot compare with the proposed algorithms, which primarily address incremental localization and mapping without any prior knowledge of their structure.

Reviewer 2 Report

The paper presents a technique for building graph-supported cityspaces based on the LeCDA algorithm and a novel dataset generated from a moving vehicle in the city of Xanthi.

I found the work interesting. However, to be accepted for publication, the following should be addressed:

- The paper is missing a section dedicated to the related work. Some work is already presented in too much detail in the introduction. This should be moved to a separate section, while the introduction gives background information and motivates the research.

- Line 72, what are examples of "more dynamic external environments"?

- Figure 1 should present numbered steps/stages of the approach. The caption also does not mention the disparity maps that are on the figure. 

- Section 3 should start with a brief introduction to what will be presented later on.

- Section 3.1 focuses on the evaluation set up, rather than the actual dataset.

- Will the dataset be publicly available? How does it look like? Will it be publicly available?

- What do the authors mean by "semantic" in the paper? Currently, there are no semantics (RDF triples, ontologies)  described. A graph is semantic if it has URIs and follows linked data principles. It should be clearly explained what is meant in the context of the work and if only a property graph is used.

- In the discussion, it would be good to mention the environmental conditions and road conditions and how they might affect the generated cityspaces. Further, to mention the current limitations of the approach. The conclusions should summarise and guide to future work.

These publications might be of interest:

Tempelmeier N, Feuerhake U, Wage O, Demidova E. Mining Topological Dependencies of Recurrent Congestion in Road Networks. ISPRS International Journal of Geo-Information. 2021; 10(4):248. https://doi.org/10.3390/ijgi10040248

Demidova, E., Dsouza, A., Gottschalk, S., Tempelmeier, N., & Yu, R. (2022). Creating knowledge graphs for geographic data on the web. ACM SIGWEB Newsletter, (Winter), 1-8.

Author Response

Dear Associate Editor and Reviewers,

With this cover letter, we are submitting the revised version of our manuscript entitled "Semantic communities from graph-inspired visual representations of cityscapes".

Thank you for the thorough review you have ensured for our manuscript and the valuable feedback provided. Please know that we have addressed all the raised comments and revised the manuscript accordingly. At this point, we wish to thank the anonymous referees for their constructive criticism, which allowed us to improve the quality of our paper. For this letter to be self-contained, we list each comment made by the reviewers and the associate editor, along with our specific responses. In addition, for your convenience, all significant changes have been highlighted in the revised manuscript with red color.

Yours sincerely,

Dr. Vasiliki Balaska

Reviewer \#2

\RC The paper presents a technique for building graph-supported cityspaces based on the LeCDA algorithm and a novel dataset generated from a moving vehicle in the city of Xanthi.

I found the work interesting. However, to be accepted for publication, the following should be addressed:

 \RC The paper is missing a section dedicated to the related work. Some work is already presented in too much detail in the introduction. This should be moved to a separate section, while the introduction gives background information and motivates the research.

\AR We acknowledge the reviewer's comment and we created different sections for the Introduction and the Related Work in the revised manuscript.

Manuscript reference: New Introduction and Related Literature sections.

\RC Line 72, what are examples of ``more dynamic external environments''?

\AR In line 72 (first manuscript version) as ``more dynamic external environments,'' we refer mainly to robot missions on urban scenes, where there are many dynamic entities such as pedestrians, moving vehicles, traffic lights, changing terrain, etc. The above sentence was revised accordingly in the updated manuscript.

Manuscript reference: In Line 92-93 of the revised manuscript.

\RC Figure 1 should present numbered steps/stages of the approach. The caption also does not mention the disparity maps that are on the figure.

\AR We thank the reviewer for his/her observation. All the details are included in the revised manuscript.

Manuscript reference: In Figure 1 and its caption.

\RC Section 3 should start with a brief introduction to what will be presented later on.

\AR In the revised version of this section (Section 4 in the updated manuscript), we have added a short introduction to what will be presented later.

Manuscript reference: In Section 4 of the revised manuscript.

\RC Section 3.1 focuses on the evaluation set up, rather than the actual dataset.

\AR We believe that the original section's title was indeed misleading. Our intent is to describe the sensors and the procedure according to which our novel dataset was formulated. In order to improve this section, its title is altered and further details are provided regarding the resulting dataset (e.g., frame rate, image resolution, sequences of different lighting conditions ). The frame rate of the camera is defined as 48fps. This dataset has 8868 unlabeled images divided in 3 different hours and 291 semantic labeled images randomly selected from the dataset. Each image file is 640 x 480 pixels and is linked with geolocation details. As in the original manuscript version, we also provide the link:\\

https://robotics.pme.duth.gr/research/gryphon\_urban,\\where everyone can download our dataset and read further specifications.

Manuscript reference: In Section 4.1 of the revised manuscript.

\RC Will the dataset be publicly available? How does it look like? Will it be publicly available?

\AR Section 3.1 of our original manuscript offered an accessible link, to our Gryphon_{urban} dataset: \\ https://robotics.pme.duth.gr/research/gryphon\_urban. \\

This is kept and highlighted in the updated version.

Manuscript reference: In Section 4.1 of the revised manuscript.

\RC What do the authors mean by "semantic" in the paper? Currently, there are no semantics (RDF triples, ontologies)  described. A graph is semantic if it has URIs and follows linked data principles. It should be clearly explained what is meant in the context of the work and if only a property graph is used.

\AR We understand the reviewer's concern and want to clarify that the context of semantics in our work refers to the semantic segmentation of the input frames. Specifically, our intent is to describe a graph of semantic entities. Our main contribution is the creation of graph-based description vectors, for the computation of which we semantically segment every input image and produce the corresponding descriptor in the form of an undirected graph. As the robot explores an environment, each scene is described through metric and semantic information. Specifically, each image captured from the robot's path is interpreted as an undirected graph. Therefore, for each semantic entity in the scene, we compute its centroid in the 3D world (cme). Graph nodes represent semantic entities (cme), while edges are weighted by the Euclidean distance between them.

Manuscript reference: The above points are clarified in Section 3.1.

\RC In the discussion, it would be good to mention the environmental conditions and road conditions and how they might affect the generated cityspaces. Further, to mention the current limitations of the approach. The conclusions should summarise and guide to future work.

\AR We embrace the reviewer's advice, so we mention the environmental and road conditions in the revised version of our work, as well as how they might affect the generated cityspaces. Furthermore, we describe some limitations of the approach. In the conclusions, we additionally highlight our plans for future work.\\

Manuscript reference: In Discussion and in Conclusions sections.

\RC These publications might be of interest:

Tempelmeier N, Feuerhake U, Wage O, Demidova E. Mining Topological Dependencies of Recurrent Congestion in Road Networks. ISPRS International Journal of Geo-Information. 2021; 10(4):248. https://doi.org/10.3390/ijgi10040248

Demidova, E., Dsouza, A., Gottschalk, S., Tempelmeier, N., \& Yu, R. (2022). Creating knowledge graphs for geographic data on the web. ACM SIGWEB Newsletter, (Winter), 1-8.

\AR The above papers were added to the Related Literature.\\

Manuscript reference: In Section 2 of the updated manuscript.

Round 2

Reviewer 1 Report

Authors did a good job explaining the novelty of the submitted paper and highlight the original contribution. All of my comments on the original article have been addressed. Reviewer appreciates the effort of the authors. There are still some minor typos, the corrections should be done before submitting the final version of the article for publishing (e.g., in keywords, “urban” should be all sub-script for the word “Gryphon_{urban}”). The revised version has improved much, and the reviewer does not have any further comments.

Author Response

Dear Associate Editor and Reviewers,

With this cover letter, we are submitting the revised version of our manuscript entitled "Semantic communities from graph-inspired visual representations of cityscapes".

Thank you for the thorough review you have ensured for our manuscript and the valuable feedback provided. Please know that we have addressed all the raised comments and revised the manuscript accordingly. At this point, we wish to thank the anonymous referees for their constructive criticism, which allowed us to improve the quality of our paper. For this letter to be self-contained, we list each comment made by the reviewers and the associate editor, along with our specific responses. In addition, for your convenience, all significant changes have been highlighted in the new revised manuscript with blue color. Note that the sentences colored red are changes from the first round review.

Yours sincerely,

Dr. Vasiliki Balaska

Reviewer 1
RC Authors did a good job explaining the novelty of the submitted paper and highlight the original contribution. All of my comments on the original article have been addressed. Reviewer appreciates the effort of the authors. There are still some minor typos, the corrections should be done before submitting the final version of the article for publishing (e.g., in keywords, “urban” should be all sub-script for the word Gryphon_urban. The revised version has improved much, and the reviewer does not have any further comments.

AR We acknowledge the reviewer's comment and we have corrected the typo in our new revised version.

Reviewer 2 Report

The authors have addressed my previous comments. Here are some comments about the new content of the introduction:

- What is meant by contemporary research? Examples should be referenced.

- What are examples of autonomous operations the authors refer to on line 22?

- What do the authors mean by the sentence on lines 25-28? AI needs semantics to become context-aware. Further, the grammar of the sentence needs to be improved.

- The introduction needs to clearly state what is lacking currently in the field and why the proposed work is important. 

Author Response

Dear Associate Editor and Reviewers,

With this cover letter, we are submitting the revised version of our manuscript entitled "Semantic communities from graph-inspired visual representations of cityscapes".

Thank you for the thorough review you have ensured for our manuscript and the valuable feedback provided. Please know that we have addressed all the raised comments and revised the manuscript accordingly. At this point, we wish to thank the anonymous referees for their constructive criticism, which allowed us to improve the quality of our paper. For this letter to be self-contained, we list each comment made by the reviewers and the associate editor, along with our specific responses. In addition, for your convenience, all significant changes have been highlighted in the new revised manuscript with blue color. Note that the sentences colored red are changes from the first round review.

Yours sincerely,

Dr. Vasiliki Balaska

Reviewer 2

RC The authors have addressed my previous comments. Here are some comments about the new content of the introduction:

RC What is meant by contemporary research? Examples should be referenced.

AR We added some citations with state-of-the-art methodologies on semantic mapping.

Manuscript reference: In lines 16-17 of the newly revised manuscript.

RC What are examples of autonomous operations the authors refer to on line 22?

AR We realise that the sentence in line 22 (first revised paper) caused ambiguity, making it difficult to understand. We revised this sentence and added some citations.

Manuscript reference: In lines 17-24 of the newly revised manuscript.

RC What do the authors mean by the sentence on lines 25-28? AI needs semantics to become context-aware. Further, the grammar of the sentence needs to be improved.

AR Due to the grammar and syntax errors, we understand the difficulty of understanding. We improved the sentence by clarifying its meaning.

Manuscript reference: In lines 25-34 of the newly revised manuscript.

RC The introduction needs to clearly state what is lacking currently in the field and why the proposed work is important. 

AR We acknowledge the reviewer's comment, so we have described in this revised manuscript what is currently lacking in the field and why the proposed work is important.

Manuscript reference: In lines 35-37 of the newly revised manuscript.